

# A model for simultaneous evaluation of NO₂, O₃ and PM₁₀ pollution in urban and rural areas: handling incomplete data sets with multivariate curve resolution analysis

Eva Gorrochategui[1,3*], Isabel Hernandez[2], Romà Tauler[3,*]

[1] University of Rennes, Inserm, EHESP, Irset (Institut de Recherche en Santé, Environnement et Travail)—UMR_S 1085, F-35000 Rennes, France.

[2]Direcció General de Qualitat Ambiental i Canvi Climàtic, Generalitat de Catalunya, Spain

[3] Institute of Environmental Assessment and Water Research (IDAEA), Consejo Superior de Investigaciones Científicas (CSIC), Barcelona, 08034, Spain.

*Correspondence to*: Eva Gorrochategui (eva.gorrochateguimatas@ehesp.fr) and Romà Tauler (rtaqam@cid.csic.es)

**Abstract.** A powerful methodology, based on multivariate curve resolution alternating least squares (MCR-ALS) with quadrilinearity constraints, is proposed to handle complex and incomplete four-way atmospheric data sets, providing concise and easy interpretable results. Changes in air quality by nitrogen dioxide (NO₂), ozone (O₃) and particular matter (PM₁₀) in eight sampling stations located in Barcelona metropolitan area and other parts of Catalonia during the COVID-19 lockdown (2020) with respect to previous years (2018 and 2019) are investigated using such methodology. MCR-ALS simultaneous analysis of the 3 contaminants among the 8 stations and for the 3 years allows the evaluation of potential correlations among the pollutants even when having missing data blocks. NO₂ and PM₁₀ show correlated profiles due to similar pollution sources (traffic and industry), evidencing a decrease in 2019 and 2020 due to traffic restriction policies and COVID-19 lockdown, especially noticeable in the most transited urban areas (*i.e.*, Vall d'Hebron, Granollers and Gràcia). Ozone evidences an opposed inter-annual trend, showing higher amounts in 2019 and 2020 respect to 2018 due to the decreased titration effect, more significant in rural areas (Begur) and in the control site (Obserbatori Fabra).

**Keywords:** Multivariate curve resolution; MCR-ALS; Multilinear modeling; incomplete data; NO₂; O₃; PM₁₀; COVID-19; lockdown.




## 1. Introduction

Monitoring studies of air quality have always been indispensable to assess the impact of air pollutants on human health and the environment. Most evaluated air pollutants include the ones linked to industrial and traffic emissions, such as tropospheric ozone ($O_3$), nitrogen dioxide ($NO_2$) and particulate matter (PM), due to its potential effects on human health (Zú et al., n.d.; Khaniabadi et al., n.d.), and are the chemicals evaluated in the present study.

The chemistry of nitrogen oxides ($NO_x$) and $O_3$ is highly complex because $NO_x$ is the responsible for tropospheric $O_3$ production but also for its elimination (Lerdau et al., 2000; Crutzen, 1979). On the one hand, the formation of tropospheric ozone is a consequence of the photochemical reaction of the sunlight with $NO_x$ and volatile organic compounds (VOC) released by car exhausts and industries, according to the following equation: $NO_x + VOC + hv \rightarrow O_3$. Thus, nitrogen oxides behave as catalysts in the photochemical production of ozone, especially at higher solar radiation and during hours of high traffic. However, at hours of low solar radiation and during nighttime, $NO_x$ are responsible for the ozone destruction in a process called titration: $NO_x + O_3 \rightarrow NO_x + O_2$. In inner rural areas with low anthropogenic activities, the latter titration effect produced by $NO_x$ emissions is generally not observed, resulting in higher average $O_3$ concentrations than in urban areas. Overall, the complex equilibrium among $O_3$ and $NO_x$ species results in continuous concentration changes of ozone difficult to attribute to a unique source.

Conversely, the chemistry of particulate matter is not directly correlated to $NO_x$ and $O_3$ but it is also complex due to its multiple and diverse emission sources. Different PM sources exist including city background (background levels of emissions such as construction, demolition and domestic heating), traffic (motor emissions and tire, pavement and brakes abrasion products), industry (high levels of sulfate, nitrate and other burning products), and natural (*i.e.*, marine aerosols and air masses, especially African dust) (Querol et al., 2004; Saud et al., 2011).

Different approaches exist to assess air quality by evaluating concentration changes of these chemical pollutants. In classical air quality monitoring studies, the data treatment strategy generally involves data arrangement and analysis using traditional statistics. However, these methods require extensive computer calculations and their results are often limited and restricted. Instead, chemometric methods are powerful data analysis tools to investigate the sources of data variance in experimentally measured environmental monitoring big data sets such as air quality data sets that often contain some missing blocks. These methods can be used to extract and summarize the information often hidden on these environmental big data sets. Among these methods, Multivariate Curve Resolution Alternating Least Squares (MCR-ALS) (Tauler, 1995), originally used in the spectrochemical analysis of chemical mixtures, has been also proved to be a competitive method in air pollution studies (Malik and Tauler, 2013; Alier et al., 2011). MCR-ALS is a flexible soft-modelling factor analysis method that allows for the introduction of natural constraints, like non-negativity of the factor solutions. Although it only requires the fulfillment of a bilinear model for the factor decomposition, it can be easily adjusted to the analysis of more complex multiway data structures and multilinear models, such as three-way and four-way environmental data sets (Tauler, 2021), which can be analyzed using trilinear and quadrilinear MCR-ALS models, as shown in this study. The results of the application of the MCR-ALS method can be used for the discovery of the main driving factors (latent



variables) responsible of the observed data variance, in this case, of the observed changes in the measured
chemical pollutants.
The present study is focused on promoting and extending the use of multivariate curve resolution alternating
least squares method, including trilinear and quadrilinear constraints, for the investigation of $NO_2$, $O_3$ and
$PM_{10}$ air pollution. In addition, this study aims at providing different strategies to deal and estimate missing
data also using the MCR-ALS methodology (Multivariate Curve Resolution of incomplete data multisets |
Elsevier Enhanced Reader, 2022). The selected chemometric strategy is ultimately used to evaluate the
temporal patterns of the three pollutants during 2018, 2019 and 2020 in eight monitoring stations located
in Catalonia (Spain), including three urban, one control site, one semi-urban and three rural. The different
stations were specifically selected to evaluate the influence of the geographical location on air pollution.
The period of time evaluated (*i.e.*, January 1st to December 31st of 2018, 2019 and 2020) was chosen to
cover the COVID-19 lockdown period in Catalonia and to enable a comparison respect to the same time
period in the previous two years. Considering that the strictest COVID-19 lockdown in Catalonia occurred
in the month of April 2020, a specific evaluation of air quality changes produced during this period of time
respect to the previous two years is provided in this study.
**2. Materials and methods**
**2.1 Air quality data**
The experimental data used in this work consisted on $O_3$, $NO_2$ and $PM_{10}$ concentrations recorded from eight
air quality monitoring stations operated by the Department of Environment of the Catalan Autonomous
Government. The selected air quality monitoring stations consisted in three urban (Gràcia, Vall d'Hebron
and Granollers), one semi-urban (Manlleu) and one control site (Observatori Fabra), all of them located in
the province of Barcelona, and three rural: Juneda and Bellver de Cerdanya in the province of Lleida, and
Begur (Costa Brava, NE Catalonia), in the province of Girona. More detailed information about the
characteristics of the stations is provided in a previous air quality monitoring study by the authors
(Gorrochategui et al., 2021). $NO_2$ concentrations were measured by means of chemiluminescence according
to the UNE method 77212:1993, using automatically operated MCV 30QL analyzers. Ozone concentrations
were measured by means of UV photometry according to ISO FDIS 139464:1998, automatically operated
with MCV 48 AUV analyzers. $PM_{10}$ concentrations were measured by means of gravimetric determination,
using manually operated high volume samplers MCV CAV-A/MS. The generated databases with all the
concentrations measured were compiled by the Department of Air Monitoring and Control Service of the
Generalitat de Catalunya (Xarxa de Vigilància i Previsió de la Contaminació Atmosfèrica (XVPCA).
Departament de Territori i Sostenibilitat, 2020).
**2.2. Experimental data multisets**
In this study, two experimental data multisets were analyzed (see Fig. 1). Both of them contained hourly
concentrations of $NO_2$, $O_3$ and $PM_{10}$ measured in the eight air quality monitoring stations but for distinct
periods of time. The first data multiset contained air quality data recorded in the month of April 2018, April
2019 and April 2020 (*i.e.*, the latter being the time when the strictest lockdown occurred in Catalonia(Real





de crisis sanitaria ocasionada por el COVID-19., 2020)) in the different stations. The second data set
contained air quality data recorded in the same 8 stations but during a longer period of time: from January
$1^{st}$ to December $31^{st}$ of 2018, 2019 and 2020. The latter multiset was built in order to evaluate annual trends
of air pollution; especially interesting in 2020, an extraordinary year due to the coronavirus pandemic.
As observed in Fig. 1, both data sets contained some missing data blocks, which were not included in the
MCR-ALS analyses of individual contaminants, a part from some spot values, which were further estimated
to undergo chemometric analysis.
In the data set of the month of April, no missing data existed for $NO_2$ and $O_3$. However, for $PM_{10}$, data of
three months of April were missing (*i.e.*, Begur 2018, Begur 2020 and Observatori Fabra 2018), as observed
in Fig. 1a. In the data set of the entire three years (Fig. 1b), for $NO_2$ and $O_3$, the months of January and
February 2018 in Observatori Fabra station were missing, respectively. For $PM_{10}$, data from three air quality
monitoring stations were missing: Gràcia (September and October, 2018), Begur (months from January to
October, 2018, and months from January to July, 2020) and Observatori Fabra (months from January to
September, 2018).

**2.3. Data sets arrangement**
In this study, the two data multisets were separately arranged to further undergo MCR-ALS individual
analyses of the complete experimental data sets (Fig. 1).
To conduct the analysis of the month of April, data matrices for $NO_2$, $O_3$ and $PM_{10}$ were separately arranged
in a first step. For each contaminant, a total of 24 data matrices, one per year (three years) and per
monitoring station (eight stations), of size 30 x 24 (month days' x hourly measurements), were obtained.
As observed in Fig. 1a these 24 data matrices were labeled as $\mathbf{D}_{station-year}$; with the name of the corresponding
air quality station (V: Vall d'Hebron, Gn: Ganollers, M: Manlleu, J: Juneda, Bl: Bellver, Ga: Gràcia, Bg:
Begur and O: Observatori Fabra) and the two last digits of the year (2018, 2019 and 2020). These 24 data
matrices were then arranged using a column-wise augmentation, obtaining three augmented data matrices:
$\mathbf{D}_{caug-April-NO2}$, $\mathbf{D}_{caug- April-O3}$ and $\mathbf{D}_{caug-April-PM10}$. The two first augmented matrices ($NO_2$ and $O_3$) contained
concentration measures of the month of April for each station and each year folded one on top of the other,
first performing the augmentation for the 3 years (30 x 3) and then for the eight stations (30 x 3 x 8), as
shown in Fig. 1a. The resulting dimensions of these two column-wise augmented data matrices for further
MCR-ALS analysis were (720 x 24). However, as previously stated, for $PM_{10}$, data of three months were
missing and thus, the final column-wise augmented matrix was built only with the six stations containing
no missing data (30 x 3 x 6), resulting in a (540 x 24) matrix (yellow-shaded area in Fig. 1a).
To conduct the analysis of the entire three years, data matrices for $NO_2$, $O_3$ and $PM_{10}$ were also separately
arranged in a second step. For each contaminant, a total of 24 data matrices, one per year and per monitoring
station, of size 365 x 24 (year days x hourly measurements), were obtained.



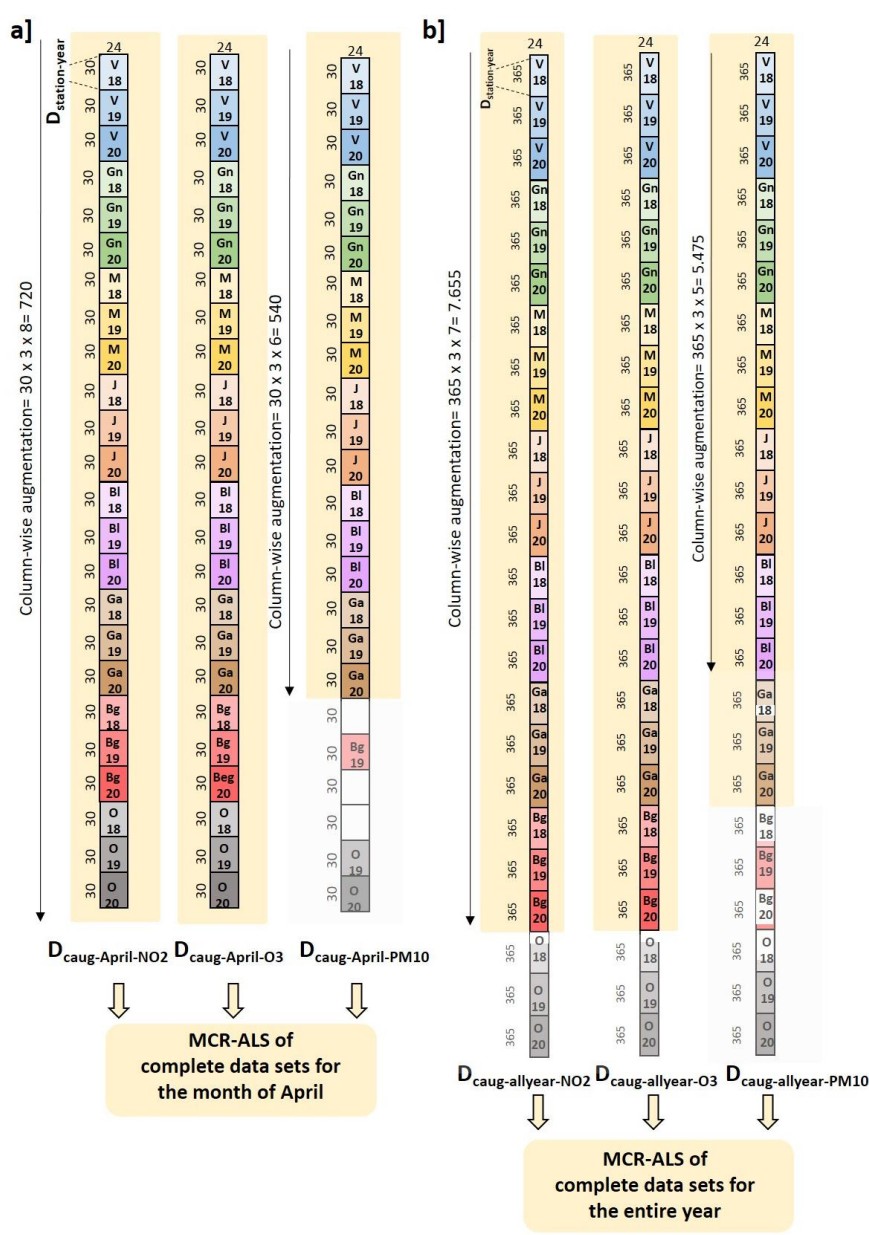

Figure 1. Data arrangement for individual analysis of completed data sets. $NO_2$, $O_3$ and $PM_{10}$ concentrations in a) the month of April and b) the entire year, for one year and one station can be arranged in a data matrix with the days in the rows and the 24 hour measurements in the columns $D_{station-April}$ or $D_{station-year}$. These individual data matrices are arranged in three (one per each pollutant) column-wise augmented data matrices. For the data set of April in 2018-2019-2020 (a), the augmented matrices are $D_{caug-April-NO2}$, $D_{caug-April-O3}$ (720,24) and $D_{caug-April-PM10}$ (540,24). For the data set of the entire year in 2018-2019-2020 (b), the augmented matrices are $D_{caug-allyear-NO2}$, $D_{caug-allyear-O3}$ (7.655,24) and $D_{caug-allyear-PM10}$ (5.475,24). Stations containing missing data (white-gaps in the figure) are excluded for further MCR-ALS analysis (yellow-shaded area). VH: Vall d'Hebron, Grn: Granollers, Mn: Manlleu, Jun: Juneda, Bell:Bellver, Gra: Gràcia, Beg: Begur, OF: Observatori Fabra.





These 24 data matrices were then arranged using a column-wise arrangement, obtaining three augmented
data matrices: $\mathbf{D_{caug\text{-}allyear\text{-}NO2}}$, $\mathbf{D_{caug\text{-}allyear\text{-}O3}}$ and $\mathbf{D_{caug\text{-}allyear\text{-}PM10}}$ (Fig. 1b). In this case, for the three
contaminants, some data were missing (white gaps in the figure). For $NO_2$ and $O_3$, some data from
Observatori Fabra station was missing. Thus, the resulting augmented matrices, $\mathbf{D_{caug\text{-}allyear\text{-}NO2}}$, $\mathbf{D_{caug\text{-} allyear\text{-}}}$
$\mathbf{_{O3}}$, contained information of the whole year, seven stations and the three years (365 x 3 x 7), resulting in
(7655 x 24) matrices, as shown in the figure. For $PM_{10}$, data of three stations were missing (white gaps in
the figure) corresponding to Gràcia, Begur and Observatori Fabra. Thus, in order to perform the MCR-ALS
analysis, the resulting $PM_{10}$ column-wise augmented matrix only contained information of five stations
with no missing data (365 x 3 x 5), resulting in a (5475 x 24) matrix (yellow-shaded area in Fig. 1b).
Data arrangement for the simultaneous study of the three pollutants considering the whole incomplete
multiblock experimental data sets is further described in Sect. 2.7.

**2.4. Estimation of missing data**
Estimation of missing data was used for the case when failures of stations and/or malfunction of them
caused the absence of measurements for few hours or few days. In order to estimate such missing data, the
nearest-neighbor method(Peterson, 2009) (*i.e.*, knn imputation) was used. In this study, the function
*mdcheck* (*i.e.*, missing data checker and infiller) of PLS Toolbox version 8.9.1 (Eigenvector Inc., WA) was
utilized to perform the imputation. This function checks for missing data and infills them using a PCA
model imputation from distinct algorithms. In our case, three algorithms were tested consisting on 'svd'
(Singular Value Decomposition), 'NIPALS' (Nonlinear Iterative Partial Least Squares) and 'knn', the latter
providing the better estimation results in our case, and thus, the one that was finally used in this study.
It is important to mention that estimation of missing data was not performed in cases where the entire month
was missing. For those cases, the station was not included in the MCR-ALS analysis of the complete data
set. For the analysis of incomplete multiblock data sets, an especial arrangement was performed using a
particular data fusion strategy, as further explained in Sect. 2.7.

**2.5. MCR-ALS analysis of the experimental data**
Different chemometric methods have been proposed in the literature for the analysis of environmental
monitoring data. MCR-ALS is a frequently used method in spectrochemical mixture data analysis, which
can also be easily extended to the analysis of environmental source apportionment data sets (Alier et al.,
2011). MCR-ALS is a flexible soft-modelling factor analysis tool which allows for the application of natural
constraints (see below) and it can be easily adapted to the analysis of complex multiway (multimode) data
structures, such as three- and four-way environmental data sets using trilinear and quadrilinear model
constraints (De Juan et al., 1998; Smilde et al., 2004; Malik and Tauler, 2013).
The simplest application of the MCR-ALS method is based on a bilinear model that performs the factor
decomposition of a two-way data set (*i.e.* a data table or a data matrix). Eq. (1) summarizes this bilinear
model in its element-wise way, while Eq. (2) presents the same model in a matrix linear algebra format:
$$d_{i,j} = \sum_{n=1}^{N} x_{i,n} y_{j,n} + e_{i,j} \quad i=1,..I \text{ (days)}, j=1,...J \text{(hours)} \tag{1}$$



$$D = X\,Y^T + E \qquad\qquad (2)$$

In Eq. (1), the individual data values, $d_{i,j}$ elements (in this case the $O_3$, $NO_2$ or $PM_{10}$ concentration values
measured one day (i) at a particular hour (j)) are decomposed as the sum of a reduced number of
contributions (components), n=1,..N, each one of them defined by the product of two factors, $x_{i,n}$ (scores)
and $y_{j,n}$ (loadings). In addition, the term $e_{i,j}$ is the residual part of $d_{i,j}$, which cannot be explained by these N
components and accounts as experimental noise and uncertainties. In Eq. (2), the data matrix, $D$, of
dimensions IxJ is decomposed into the scores factor matrix $X$ (IxN) and the loadings factor matrix, $Y^T$
(NxJ). The number of components, N, is selected to explain as much as possible the data variance, while
the unexplained small contributions of data variance and experimental noise are in $E$. Multivariate Curve
Resolution(Tauler, 1995) performs the bilinear model factor decomposition shown in Eq. (1) and (2) using
an alternating least squares (MCR-ALS) algorithm under a set of constraints which reduce the extent of the
bilinear model rotation ambiguities(Abdollahi and Tauler, 2011)  and allow the physical identification and
interpretation of the factor matrices $X$ and $Y^T$, as for example, the application of non-negativity constraints
to the elements of the factor matrices $X$ and $Y^T$(Bro and De Jong, 1997; De Juan and Tauler, 2003). Models
with different number of components can be tested and a final decision is taken considering the data fit and
the shapes and reliability of the resolved profiles. The ALS algorithm also needs initial estimates of either
$X$ or $Y^T$ factor matrices. These initial estimates can be obtained from the more 'dissimilar' rows or columns
of the original data matrix(Jackson et al., 2002). Eq. (2) for $D$ is solved iteratively, which updates the
solutions (vector profiles in $X$ and $Y^T$ matrices) until they fit the data optimally and fulfill the proposed
constraints
In this work the MCR-ALS method has been applied either to the individual data matrices $D_{station,year}$
described in previous section and in Fig. 1 for every pollutant ($O_3$, $NO_2$ or $PM_{10}$), at one period of time
(April or full year) and at one monitoring station or to the augmented matrices of the same three pollutants
for the three years (k=1,…3) simultaneously and for the different stations (l=1,…,8) concatenated vertically
in $D_{caug\text{-}April}$ or $D_{caug\text{-}allyear}$ (see Fig. 1). In the case of the individual data matrices described above for the
period of time (April or full year), $D_{station,april}$ or $D_{station,year}$, the factor (scores) matrix $X$ will have
respectively the April or full year day profiles of the components and the factor (loadings) matrix $Y^T$ will
have the corresponding hour profiles of these components. In the case of the column-wise augmented data
matrices $D_{caug\text{-}April}$ or $D_{caug\text{-}allyear}$, bilinear model Eq. (2) was extended as:

$$D_{caug} = X_{caug}\,Y^T + E_{caug} \qquad\qquad (3)$$

Where $X_{caug}$ is now the augmented factor (scores) matrix with the augmented day profiles concatenated
vertically for the different years and stations, and $Y^T$ is the matrix of the hour profiles again, which are
common for all the concatenated matrices in $X_{caug}$. During the ALS optimization of the bilinear model in
Eq. (3), constraints can be also applied and the same aspects in relation to number of components and
convergence as for solving Eq. (2) are considered.
**2.6 MCR-ALS analysis of the complete experimental data sets using trilinear and quadrilinear**
**constraints**
Solving Eq. (3) using bilinear MCR-ALS does not take into account the temporal and spatial structure of





the data in the vertical concatenated mode which includes the information of the day, year and station. This
data structure can be considered in the trilinear and specially in the quadrilinear extensions of the bilinear
models described in Eq. (1-3).
The factor decomposition model given before can be extended to a three-way dataset, **$\underline{D}$,** or to a four-way
data set, **$\underline{D}$,** expressed individually for every data value as given by Eq. (4) and (5).

$d_{i,j,kl} = \sum_{n=1}^{N} x_{i,n} y_{j,n} z_{kl,n} + e_{i,j,kl}$        (trilinear model)        (4)

$d_{i,j,k,l} = \sum_{n=1}^{N} x_{i,n} y_{j,n} z_{k,} w_{l} + e_{i,j,k,l}$        (quadrilinear model)    (5)

where $d_{i,j,kl}$ are the individual data values (concentrations of $O_3$, $NO_2$ or $PM_{10}$) in the four experimental data
modes: the day of April or of the full year i=1,…30 or i=1,...,365, the hour of the day j=1,...,24, and the
year-station kl=1,….24 in the case of the trilinear model, and $d_{i,j,k,l}$ .has the year-station third mode separated
in year k=1,2,3, and station, l=1,…,8 in the quadrilinear model. These data values are modeled as the sum
of a number of components (contributions), n=1,..N, defined by the product of three factors $x_{i,n}$, $y_{j,n}$, and
$z_{kl,n}$, in the case of the trilinear model and in four factors in the case of the quadrilinear model, $x_{i,n}$, $y_{j,n}$, $z_{k,n}$
and **$w_{ln}$**. These factors are related with the three and four data modes respectively (day, hour and year-
station or day, hour, year and station). **$e_{i,j,kl}$** and **$e_{i,j,k,l}$** are the part of **$d_{i,j,kl}$** and **$d_{i,j,k,l}$** not explained by the
contribution of these N components. These trilinear and quadrilinear models can be written in a matrix form
using the decomposition of every individual **$D_{kl}$** data slice (every individual matrix **$D_{kl}$**), as shown in Eq.
(6) and (7).

$\mathbf{D_{kl} = X\ Z_{kl}\ Y^{T} + E_{kl}}$          (trilinear model)         (6)

$\mathbf{D_{kl} = X\ Z_{k}\ W_{l}\ Y^{T} + E_{kl}}$      (quadrilinear model)    (7)

Under the trilinear model, all individual data matrices, **$D_{kl}$**,(I,J) are simultaneously decomposed with the
same number of components N and the same daily, **X** (I,N) and hourly **$Y^{T}$** (N,J) profiles. Thus, they differ
only in a diagonal matrix **$Z_{kl}$**, (N,N) different for every one of the kl=1,…,24 year-stations (year-station
profiles), which gives the relative amounts of the N components in every data matrix (year-station), **$D_{kl}$**.
These N diagonal elements of the **$Z_{kl}$** can also be grouped in the third factor matrix **Z** (KxL,N). Under the
quadrilinear model, all individual data matrices, **$D_{kl}$**,(I,J) are simultaneously decomposed with the same
number of components N and the same daily, **X** (I,N) and hourly **$Y^{T}$** (N,J) profiles. Thus, they differ in the
diagonal matrices **$Z_{k}$**, (N,N) and **$W_{l}$** (N.N), which are different for every year (k) and station (l), which give
the relative amounts of the N components in every data matrix **$D_{kl}$** respectively. These N diagonal elements
of the **$Z_{k}$** and **$W_{l}$** matrices can also be grouped in the third and four factor matrices **Z** (K,N) and **W**(L,N).
Therefore, the proposed trilinear and quadrilinear models take advantage of the natural structure of the
analyzed data sets, especially in relation to their different temporal modes (*i.e.* hourly, daily, yearly) and to
the different type of monitoring stations analyzed simultaneously. The implementation of trilinear and
quadrilinear models as a constraint in the MCR-ALS method has been described in previous works(Tauler,
2021; Malik and Tauler, 2013; Alier et al., 2011) Here only a brief explanation of the case of the
implementation of the quadrilinear model constraint for the case of study is shown.





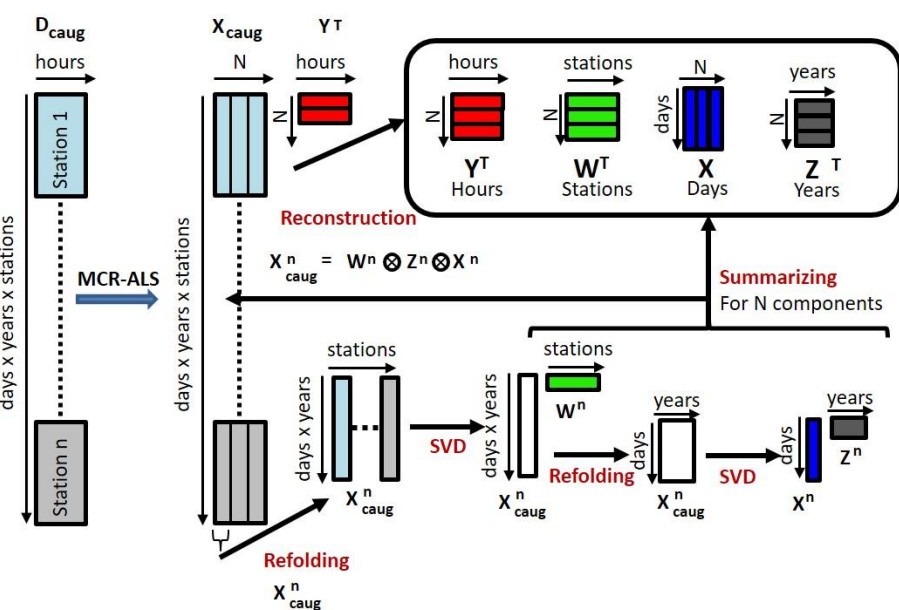


**Figure 2. MCR-ALS with the quadrilinearity constraint. Graphical description of the implementation of the quadrinilearity constraint during the Alternating Least Squares optimization. See Eq. (4-7) and their explanation in the manuscript.**

Figure 2 shows the practical implementation of the quadrilinear constraint in the MCR-ALS analysis of the
four-way data set obtained in the two type of data, when the April data of the three parameters ($O_3$, $NO_2$
and $PM_{10}$) were studied over the three years (2018-2019 and 2020) and over the different monitoring
stations described above, and also for the analogous four-way data set when instead of April data, the full
year data were considered for the same years and stations.
The individual data sets with the concentrations of the three parameters (one per year and station), were
arranged in the column-wise augmented data matrix $\mathbf{D_{caug}}$ of dimensions 30 (April) or 365 days x 3 years
x 8 stations, giving a total number of 720 rows for April data or of 8760 rows for the full year data, and 24
hourly measures in columns. These number of row elements is for the case of no-missing data, however
they will be lowered for the cases of missing data, especially in the case of the full year data (see previous
section in missing data). The application of the quadrilinearity constraint implies that the augmented
profiles of every component n, $\mathbf{x^n_{aug}}$, having the vertically concatenated information of days times years
times stations is first refolded in the data matrix $\mathbf{X^n_{aug}}$ of dimensions 30 (April) or 365 (all year) times 3
(years) rows by 8 (stations). This augmented factor matrix is decomposed by SVD considering only the
first singular component into the product of two vector profiles, one long vector profile $\mathbf{x^n_{caug}}$ (90 or 1095
x 1) of the combined day-year profile by a vector profile $\mathbf{w^n}$ (8x1) describing differences among the
different stations for the component n. $\mathbf{x^n_{caug}}$ long vector day-year profile can be further refolded in a matrix
and decomposed by SVD into the product of two new vector profiles, one related with the year profile, $\mathbf{z^n}$,
and another with the day profile $\mathbf{x^n}$, for the considered component n. In this way, for every component



(contribution), the concentration of any one of the three parameters ($O_3$, $NO_2$ and $PM_{10}$) is decomposed in
the product of four profiles, one related with the day (of April or of the whole year), $\mathbf{x^n}$, another related with
the hour of the day, $\mathbf{y^n}$, another with the considered year, $\mathbf{z^n}$, and another with the monitoring station, $\mathbf{w^n}$.
This factor decomposition allows a detangled interpretation of the temporal and spatial sources of variation
of the observed concentrations of the three pollutants. Therefore, the application of this quadrilinearity
constraint implies that for every component, the daily changes are described by the same single $\mathbf{x^n}$ vector
profile which changes over the years and station by station by the corresponding scalars values in $\mathbf{z^n}$ and
$\mathbf{w^n}$. Once the three profiles in the three augmented modes, $\mathbf{x_n}$, $\mathbf{z_n}$ and $\mathbf{w_n}$, are obtained, they can be multiplied
using the kronecker product(Soloveychik and Trushin, 2016) to reconstruct the long vector profile, $\mathbf{x^n_{aug}}$,
(see Fig. 2) and rebuild the bilinear model in the next iteration of the general ALS optimization. Finally,
the vector profiles for every component n in the different modes, can be grouped in the corresponding factor
matrices $\mathbf{X}$, $\mathbf{Z}$ and $\mathbf{W}$, which together $\mathbf{Y^T}$ give the full quadrilinear decomposition of the four-way data set,
$\underline{\mathbf{D}}$. See previous works for a more detailed description of the algorithm used for the practical implementation
of the quadrilinearity constraint in MCR-ALS(De Juan and Tauler, n.d.; De Juan et al., 1998).

**2.7 MCR-ALS simultaneous analysis of incomplete multiblock experimental data**

The simultaneous analysis of the $NO_2$, $O_3$ and $PM_{10}$ experimental data can be done one step forward using
a data fusion multiblock strategy. This would imply building a single MCR model for the whole multiset
data obtained for the 3 pollutants, $NO_2$, $O_3$ and $PM_{10}$ in April or in the whole year, for the three years, 2018,
2019 and 2020, and for the different monitoring stations. This is expressed in the following data matrix
equation (see also Fig. 3):
$\mathbf{D_{craug}} = [\mathbf{D_{caugNO2}}, \mathbf{D_{caugO3}}, \mathbf{D_{caugPM10}}] = \mathbf{X_{caug}} [\mathbf{Y^T_{NO2}}, \mathbf{Y^T_{O3}}, \mathbf{Y^T_{PM}}] = \mathbf{X_{caug}Y^T_{raug}}$     (8)
In this Equation the column-wise augmented data matrices, $\mathbf{D_{caugNO2}}$, $\mathbf{D_{caugO3}}$, and $\mathbf{D_{caugPM10}}$ previously
described (and analyzed separately by MCR-ALS with  different factor decomposition models, bilinear,
trilinear and quadrilinear), are now concatenated horizontally giving the new single row and column-wise
super-augmented data matrix $\mathbf{D_{craug}}$ which is decomposed in the two new augmented factor matrices, $\mathbf{X_{caug}}$
and $\mathbf{Y^T_{raug}}$ using the MCR bilinear model and constraints, like it was described in Sect. 2.4. In the
augmented rows of the new $\mathbf{Y^T_{raug}}$ will be the resolved hour profiles for the three contaminants $\mathbf{Y^T_{NO2}}$, $\mathbf{Y^T_{O3}}$
and $\mathbf{Y^T_{PM}}$. In addition, if the trilinearity/quadrilinearity constraints are applied to the columns of the
resolved factor matrix $\mathbf{X_{caug}}$ as described above in Sect. 2.6 using matrix decompositions of Eq. (6) and (7),
the common day, year and station profiles will be separately recovered and analyzed.
However as previously described, April and the whole year individual data sets were not obtained for all
stations, years and pollutants, and therefore they together could not be fitted in a rectangular super-
augmented data matrix containing all the data for all the years and stations as shown in Eq. (8) for $\mathbf{D_{craug}}$.
Some of the individual data sets were missing (see Sect. 2.3 and 2.4). In particular, in the case of April, two
different data blocks could be arranged. First, the $NO_2$, $O_3$ and $PM_{10}$ concentrations data for 3 years and 6
stations were arranged in the complete row- and column-wise augmented April data block, $\mathbf{DA1_{craug}}$, with
540 rows (30 days x 3 years x 6 stations) and 72 columns (24 hours for $NO_2$ + 24 hours for $O_3$ + 24 hours
for $PM_{10}$). Secondly, the additional $NO_2$ and $O_3$ concentration data for 3 years and 2 stations were arranged



in complete row- and column-wise augmented April data block $\mathbf{DA2_{craug}}$ with 180 rows (30 days x 3 years
x 2 stations) and 48 columns (24 hours for $NO_2$ + 24 hours for $O_3$). These two April data blocks can be
analyzed independently, but a new dataset can be built concatenating the two data blocks as shown in Fig.
S1, which will be reformulated and analyzed as shown in next Equation.
$\mathbf{DA12_{craug}} = [\mathbf{DA1_{craug}}; [\mathbf{DA2_{craug}}, \mathbf{NaN}(180,24)]] = \mathbf{XA12_{caug}} \mathbf{YA12^T_{raug}} =$

$= \mathbf{XA12_{caug}} [\mathbf{Y_{NO2}}, \mathbf{Y_{O3}}, \mathbf{Y_{PM10}}]$ $\qquad$ (9)

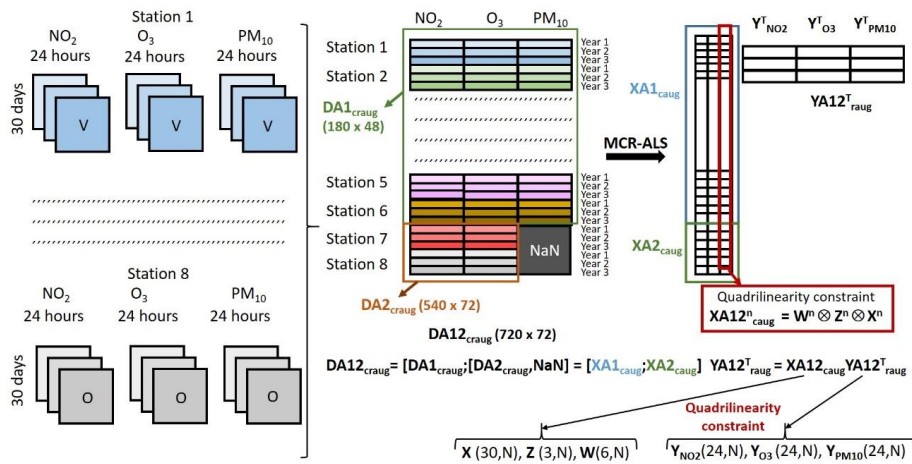


**Figure 3. MCR-ALS with the quadrilinearity constraint for the simultaneous analysis of the three contaminants in the incomplete multiblock data set of the month of April. See Eq. (9) and their explanation in the manuscript.**

This new incomplete data set $\mathbf{DA12_{craug}}$ is built using the two data blocks previously defined, $\mathbf{DA1_{craug}}$ and
$\mathbf{DA2_{craug}}$, both concatenated vertically and filling the empty data block corresponding to the unknown
concentrations of $PM_{10}$ for two missing stations with the NaN notation. The application of MCR-ALS to
this incomplete dataset is decomposed using a bilinear model (see in Fig. 3), giving the two factor matrices
$\mathbf{XA12_{caug}}$ and $\mathbf{YA12^T_{raug}}$. $\mathbf{XA12_{caug}}$ factor matrix has the column-wise augmented day x year x station
profiles in its columns and $\mathbf{YA12^T_{raug}}$ factor matrix has the row-wise augmented hour profiles for $NO_2$
($\mathbf{Y_{NO2}}$), $O_3$ ($\mathbf{Y_{O3}}$) and $PM_{10}$ ($\mathbf{Y_{PM10}}$) in its rows. As previously, the trilinear/quadrilinear constraints can be
applied during the ALS factor decomposition to the $\mathbf{XA12_{caug}}$ factor matrix and allow the separate recovery
of the day, year and station profiles, a part of the hour profiles for $NO_2$, $O_3$ and $PM_{10}$ obtained in $\mathbf{YA12^T_{raug}}$.
Analogous equations can be described for the $NO_2$, $O_3$ and $PM_{10}$ experimental data measured not only in
April but during the whole year. In this case however the two data blocks, $\mathbf{DY1_{craug}}$ and $\mathbf{DY2_{craug}}$, will have
different sizes than for the only April month data because they are for all the days of the whole year.
Different data sets were missing in this case. $\mathbf{DY1_{craug}}$ has the data for 5 stations with 5475 rows (365 days
x 3 years x 5 stations) and 72 columns (24 hours for $NO_2$ + 24 hours for $O_3$ + 24 hours for PM), and $\mathbf{DY2_{craug}}$
has the additional data for 2 stations but only for $NO_2$ and $O_3$ concentrations, with 2190 rows (365 sys x 3
years x 2 stations) and 48 columns (24 hours for $NO_2$ + 24 hours for $O_3$) (see Fig. S2). For the whole year




data, the bilinear factor decomposition can be described by the new Eq. (10).

$\mathbf{DY12_{craug}} = [\mathbf{DY1_{craug}};[\mathbf{DY2_{craug}},\mathbf{NaN}(2190,24)]] = \mathbf{XY12_{caug}}\ \mathbf{YY12^{T}_{raug}} =$

$= \mathbf{XY12_{caug}}\ [\mathbf{Y_{NO2}},\mathbf{Y_{O3}},\mathbf{Y_{PM10}}]$           (11)

Where now $\mathbf{DY12_{craug}}$ is the new incomplete data set built with the two data blocks $\mathbf{DY1_{craug}}$ and $\mathbf{DY2_{craug}}$
concatenated vertically and $\mathbf{NaN}$ (2190, 24) is for the missing $PM_{10}$ concentrations during 3 years in the
missing 2 stations (see Supplement Fig. S2). $\mathbf{XY12_{caug}}$ and $\mathbf{YY12^{T}_{raug}}$ are now the two factor matrices
obtained in the bilinear decomposition of $\mathbf{DY12_{craug}}$. The first factor matrix $\mathbf{XY12_{caug}}$ have the column-wise
augmented day x year x station profiles in its columns and the second factor matrix $\mathbf{YY12^{T}_{raug}}$ has the hour
profiles for $NO_2$, $O_3$ and $PM_{10}$ in its rows. And also as previously, the trilinear/quadrilinear constraints
applied during the ALS factor decomposition to the $\mathbf{XY12_{caug}}$ factor matrix will allow the separate recovery
of the day, year and station profiles, apart from the hour profiles for $NO_2$ ($\mathbf{Y_{NO2}}$), $O_3$ , ($\mathbf{Y_{O3}}$), and $PM_{10}$
($\mathbf{Y_{PM10}}$) obtained in $\mathbf{YY12^{T}_{raug}}$ The difference with the results of April data is that now the column-wise
augmented profiles in $\mathbf{XY12_{caug}}$ will have information about the 365 days of the whole year and not only
for the 30 days of April. Supplement Fig. S3 is given to illustrate graphically the bilinear model applied to
the incomplete two-blocks data set.

**2.8 Evaluation of MCR-ALS results**
The final evaluation of the MCR-ALS fitting results is performed calculating the explained data variances
($R^2$) using Eq. (11),

$$R^2 = 100 \times \left(1 - \frac{\sum_{i=1}^{m}\sum_{j=1}^{n}\left(d_{ij}-\hat{d}_{ij}\right)^2}{\sum_{i=1}^{m}\sum_{j=1}^{n}d_{ij}^2}\right)$$          (11)

where $\mathbf{d}_{ij}$ are the experimental predicted $O_3$ $NO_2$ or $PM_{10}$ concentrations, and $\hat{d}_{ij}$ are the corresponding
calculated values by MCR-ALS using either the bilinear (Eq. (1-3)), trilinear (Eq. (4) and (6)) or
quadrilinear (Eq. (5) and (7)) models. Apart from the global fitting with the full model (all components),
the explained variances can be also calculated individually for every MCR-ALS component, where now
the calculated values, $\hat{d}_{ij}$, take only into account one of the n components of the model. In this way, the
relative importance of the different contributions can be evaluated, as well as their overlapping degree with
the other contributions or components.
**2.8 Software**
MATLAB 9.10.0 R2021a (The MathWorks, Inc., Natick, MA, USA) was used as the development platform
for data analysis and visualization. The new graphical interface MCR-ALS GUI 2.0(Malik and Tauler,
2013), freely available as a toolbox at the web address http://www.mcrals.info/., was used for bilinear and
trilinear data sets. Statistics Toolbox™ for MATLAB and PLS Toolbox 8.9.1 (Eigenvector Research Inc.,
Wenatchee, WA, USA) were also used in this work. New specific MCR-ALS command line code for
incomplete multiblock data sets is under final development and it can be requested to one of the authors
(RT, email:roma.tauler@idaea.csic.es).



**3. Results and discussion**
Results of MCR-ALS will be shown separately for the analysis of the month of April and for the analysis
of the entire years. In the study of the month of April, the individual analysis of the three contaminants per
separate is firstly performed, using only data from stations with no missing blocks (*i.e.*, data matrices $\mathbf{D_{caug-}}$
$\mathbf{_{April-NO2}}$, $\mathbf{D_{caug-April-O3}}$ and $\mathbf{D_{caug-April-PM10}}$, yellow-shaded area of Fig. 1a). Then, a simultaneous analysis of
the three contaminants containing incomplete data is performed (*i.e.*, data matrix $\mathbf{D_{A12craug}}$, Fig. S1). In the
study of the entire years, again the individual analysis of the three contaminants per separate is initially
performed, using only data from stations with no missing blocks (*i.e.*, data matrices $\mathbf{D_{caug-allyear-NO2}}$, $\mathbf{D_{caug-}}$
$\mathbf{_{allyear-O3}}$ and $\mathbf{D_{caug-allyear-PM10}}$, yellow-shaded area of Fig. 1b). Then, a simultaneous analysis of the three
contaminants containing incomplete data is performed (*i.e.*, data matrix $\mathbf{D_{Y12craug}}$, Fig. S3). In all cases the
selection of the number of components and the initial estimates for MCR-ALS were performed as described
in Sect. 2.5. A summary of the explained variances of the MCR-ALS analyses for the different data sets
with non-negativity and either bilinear, tri-linear or quadrilinear modeling and with different number of
components is given in **Table 1**.
**Table 1.** MCR-ALS decomposition and explained variances for the different models.

| | Explained variances: April 2018-2019-2020. | | |
| --- | --- | --- | --- |
| | MCR-ALS biliineal | MCR-ALS quadrilineal | MCR-ALS trilineal |
| $\mathbf{D_{caug-April-NO2}}$[d] | All 94.4% | All 78.4% | All 79.2% |
| (4 comp) | Sum 125.2% | Sum 109.8% | Sum 113.5% |
| $\mathbf{D_{caug- April-O3}}$[d] | All 98.4% | All 92.9% | All 93.5% |
| (3 comp) | Sum 143.5% | Sum 118.3% | Sum 126.4% |
| $\mathbf{D_{caug- April-PM10}}$[d] | All 91.8% | All 78.4% | All 79.0% |
| (3 comp) | Sum 126.5% | Sum = 112.9 | Sum= 113.0% |
| $\mathbf{DA12_{craug}}$[d] | All 96.2% | All 90.7% | All 91.2% |
| (5 comp) | Sum 135.5% | Sum 111.3% | Sum 114.4 |
| | Explained variances: All year 2018-2019-2020. | | |
| $\mathbf{D_{caug-allyear-NO2}}$[d] | All 95.1.0% | All 80.3% | All 80.5% |
| (4 comp) | Sum 131.0% | Sum 116.2% | Sum 115.7% |
| $\mathbf{D_{caug- allyear-O3}}$[d] | All 97.5% | All 90.1% | All 90.6% |
| (3 comp) | Sum 132.2% | Sum 130.2% | Sum 129.2% |
| $\mathbf{D_{caug- allyear-PM3}}$[d] | All 88.1% | All 72.4% | All 72.6% |
| (3 comp) | Sum 116.6% | Sum 105.0% | Sum 103.2% |
| $\mathbf{DY12_{craug}}$[d] | All 94.7% | All 86.4% | All 86.8% |
| (5 comp) | Sum 126.8% | Sum 125.8% | Sum 126.6% |

[a] MCR-ALS for raw data with non-negativity constraint
[b] MCR-ALS for raw data with non-negativity and quadrilinear constraint
[c] MCR-ALS for raw data with non-negativity and trilinear constraint
[d] Augmented data matrices and number of components (see Fig. 1, Eq. (3), (9) and (10), and explanation in section Data sets arrangement)

MCR-ALS bilinear analysis of April data in the $\mathbf{D_{caug-April-NO2}}$, $\mathbf{D_{caug-April-O3}}$ and $\mathbf{D_{caug-April-PM10}}$ data matrices
with non-negativity constraints explained respectively 94.40%, 98.4% and 91.8% of the total variance when
four, three and three components were considered (Table 1). These values indicate the higher complexity
of the NO$_2$ data compared to O$_3$ data as will be shown also below. When the quadrilinear constraint was
applied these values decreased to 78.4%, 92.9% and 78. 4% respectively, confirming again the less complex
and more regular changes of ozone concentrations in the three years at the different monitoring stations.
Variances explained by the individual components are given in the Figures shown below. The amount of





variance overlap (also given in Table 1) in every case can be obtained subtracting the sum of the individual
variances with the variance obtained with all the components simultaneously. This difference is again larger
in the case of $NO_2$. In Table 1 also, the variances obtained when the trilinearity constraint was applied,
instead of the quadrilinearity constrain, are also given, with similar results to those obtained by both
multilinear models. In the case of MCR-ALS of all-year data of the $\mathbf{D_{caug\text{-}allyear\text{-}NO2}}$, $\mathbf{D_{caug\text{-}allyear\text{-}O3}}$ and $\mathbf{D_{caug\text{-}}}$
$\mathbf{_{allyear\text{-}PM10}}$ data matrices, rather similar results to those from April were obtained in terms of explained
variances for all three type of models (see Table 1), reflecting again the higher complexity of the NO2 data
over the years and stations compared to O3 data, and the intermediate behavior of PM10 data, although the
later more similar to the $NO_2$ data.
Possible correlations between $NO_2$, $O_3$ and $PM_{10}$ data sets during the month of April of 2018, 2019 and
2020 and in the eight stations were investigated using the incomplete data arrangement described in Sect.
2.7 and Fig. S1. MCR-ALS analysis of $\mathbf{DA12_{craug}}$ with five components and with only negativity constraints
gave a total explained variance of 96.2% (Table 1). When the quadrilinearity constraint was applied, the
total explained variance decreased down to 90.7% (91.2% for the MCR-ALS trilinear). Such decrease
between bilinear and quadrilinear MCR-ALS models of only 5 % indicated a good quadrilinear behavior
of the whole system in April. The explained variances of each component individually are given below
with the corresponding Figures of the resolved profiles. In the case of the simultaneous study of $NO_2$, $O_3$
and $PM_{10}$ profiles along all the three years (*i.e.*, 2018, 2019 and 2020) in the seven stations using the
incomplete data arrangement described in Sect. 2.7 and Fig. S3. Results using five components indicated
also a rather good quadrilinear behavior of the system. A more detailed description of the profiles describing
the concentration changes of the three pollutants and of the behavior of the whole systems formed by all of
them in the different stations and during the three years, separately for April and for the all year, is given
below.

**3.1. Study of the month of April**
**3.1.1. $NO_2$ study ($D_{caug\text{-}April\text{-}NO2}$ data matrix)**
In Fig. 4, from left to right, the profiles of the different modes of the four components are shown: x-day
(blue), z-year (black), w-station (green) and y-hours (red). Component profiles in the four modes obtained
by MCR-ALS when using non-negativity and quadrilinearity constraints are shown in Fig. 4.
$NO_2$ hour profile of first component (C1) showed a narrow maximum between 9:00-11:00 hours, coincident
with the rush traffic hour and due to fuel combustion by vehicles. In second component (C2) this hour
profile presented a much wider peak during daily hours (10:00-20:00h), again potentially attributed to the
combined effects of traffic emissions and ozone formation (see below). Third component (C3) reached an
hourly maximum in the late evening, approximately at 22:00 hours whereas forth component (C4) showed
a maximum between 00:00 and 05:00 hours, describing the $NO_2$ night behavior. As observed in the year
profiles (z-mode), for all the components, $NO_2$ contributions showed a significant decrease in 2019 and
even higher in 2020 respect to 2018; the latter possibly attributed to the COVID-19 curfew and mobility
restrictions. Moreover, as observed in the station profiles (w-mode), such depletion was consequently more
notorious in the three urban stations Vall d'Hebron (1), Granollers (2) and Gràcia (6), which were the




stations with higher $NO_2$ concentration levels. Considering that the principal emission source of $NO_2$ is
traffic, it is reasonable that the four MCR-ALS resolved components evidenced a decline in the year-mode,
corresponding to a diminution in April 2020 (under the strictest lockdown), compared to 2019 (under no
pandemia) and to 2018 (under no pandemia and no other traffic restrictions in Barcelona, such as the low
emission zones(LEZ - Àrea Metropolitana de Barcelona, 2020)). Moreover, as stated in a previous study of
the authors(Gorrochategui et al., 2021), in April 2020 an historical record of rainfall was registered in the
control site of Observatory Fabra. Therefore, the highly rainy conditions of April 2020 favored the
cleansing of the atmosphere, including $NO_2$ gases. Finally, the day profiles (x-mode) for the different
components did not show any particular pattern for the different days of the month.

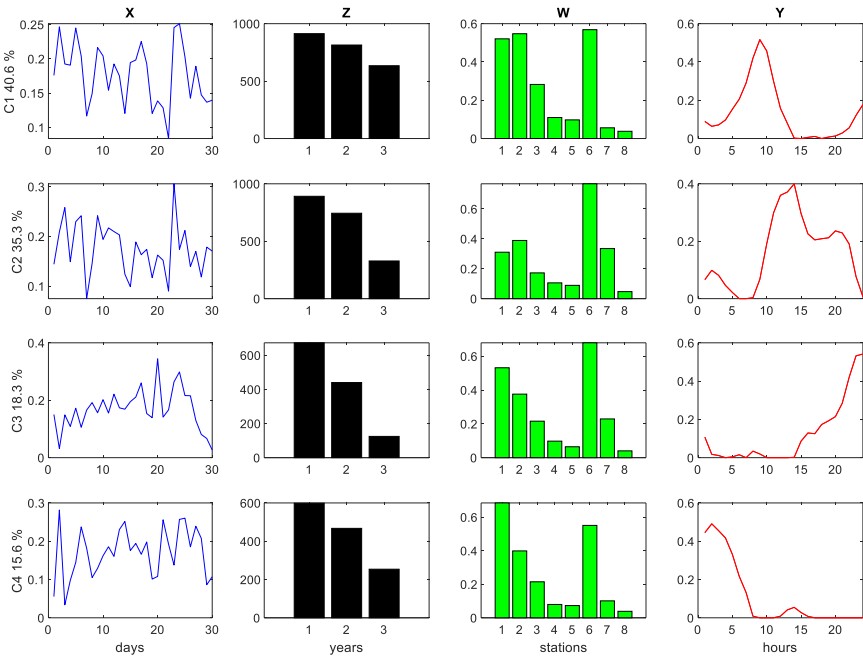


**Figure 4. MCR-ALS analysis of $NO_2$ concentrations in the column-wise super-augmented data matrix $D_{caug-April-NO2}$ (Eq. (3)) using non-negativity and quadrilinearity constraints. Profiles of the four different data modes are given in different colors: (X) in blue days of April; (Z) in black year 1=2018, 2=2019 and 3= 2020; (W) in green stations 1: Vall d'Hebron, 2: Granollers, 3: Manlleu, 4: Juneda, 5: Bellver, 6: Gràcia, 7: Begur, 8: Observatori Fabra; and (Y) in red, hours of the day.**
**3.1.2. $O_3$ study ($D_{caug-April-O3}$ data matrix)**
Profiles obtained by MCR-ALS for the three components using non-negativity and quadrilinearity
constraints are shown in Fig. 5. MCR-ALS hourly (y-mode) resolved profiles of the first component (C1)
showed a maximum between 14:00 and 22:00h, due to the cumulative solar radiation. There was practically
no difference on this component among stations, among years nor among the days of the month. The MCR
hourly resolved profile of the second component (C2) showed a different $O_3$ profile, corresponding to the
concentration at night. As observed in the w-mode, $O_3$ concentration at night was higher in the rural station



of Begur and in the control site Observatori Fabra, the latter emplaced in Collserola mountain and receiving
only some impact from Barcelona's city. The higher $O_3$ nightly concentration observed in these stations is
due to the fact that in inner rural areas, as well as in the control site, with low anthropogenic activities, the
titration effect (*i.e.*, ozone destruction under no solar radiation) produced by $NO_2$ emissions is generally not
observed, resulting in higher average $O_3$ concentrations than in urban areas. Finally, third component (C3)
showed again a maximum between 16:00 and 21:00 h, similar to the behavior described by C1, but narrower
and with a pattern among stations different to C1.

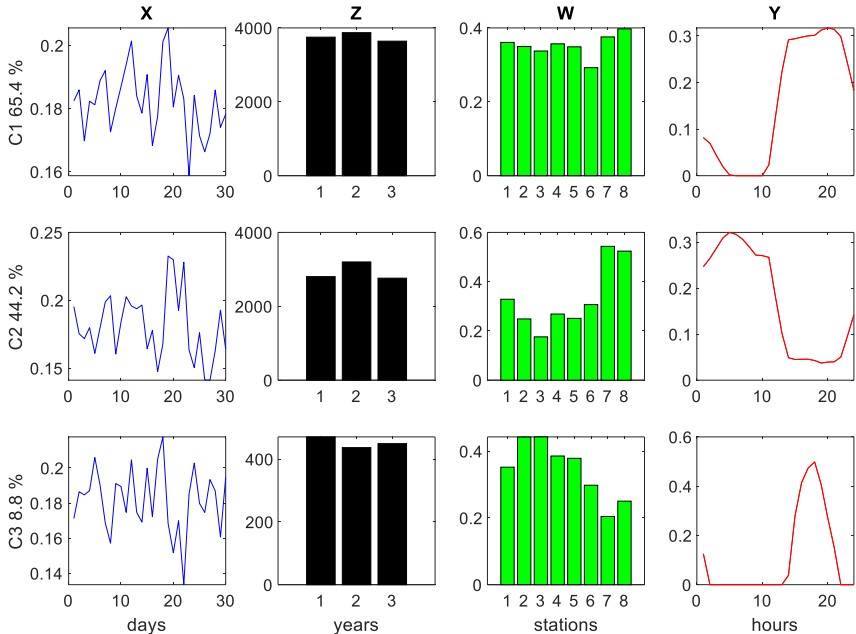


**Figure 5. MCR-ALS analysis of $O_3$ oncentrations in the column-wise super-augmented data matrix $D_{caug-April-}$**
**$_{O3}$ (Eq. (3)) using non-negativity and quadrilinearity constraints. Profiles of the four different data modes are**
**given in different colors: (X) in blue days of April; (Z) in black year 1=2018, 2=2019 and 3= 2020; (W) in green**
**stations 1: Vall d'Hebron, 2: Granollers, 3: Manlleu, 4: Juneda, 5: Bellver, 6: Gràcia, 7: Begur, 8: Observatori**
**Fabra; and (Y) in red, hours of the day.**
**3.1.3. PM$_{10}$ study ($D_{caug-April-PM10}$ data matrix)**
Profiles obtained by MCR-ALS for these three components using non-negativity and quadrilinearity
constraints are shown in Fig. 6. MCR hourly resolved profiles in the y-mode for the three resolved
components indicated a wide maximum between 00:00 and 15:00h (C1), between 15:00 and 22:00h (C2)
and between 10:00 and 20:00h (C3). As observed in the year profile (z-mode), PM$_{10}$ contribution decreased
in 2019 but most significantly in 2020, probably due to the COVID-19 lockdown. This behavior was the
same observed for $NO_2$ (Fig. 4), and it is due to the fact that among the PM$_{10}$ sources, traffic should be also
included. Moreover, such depletion was more evident in the urban stations profile (w-mode) of Vall
d'Hebron, Granollers and Gràcia.



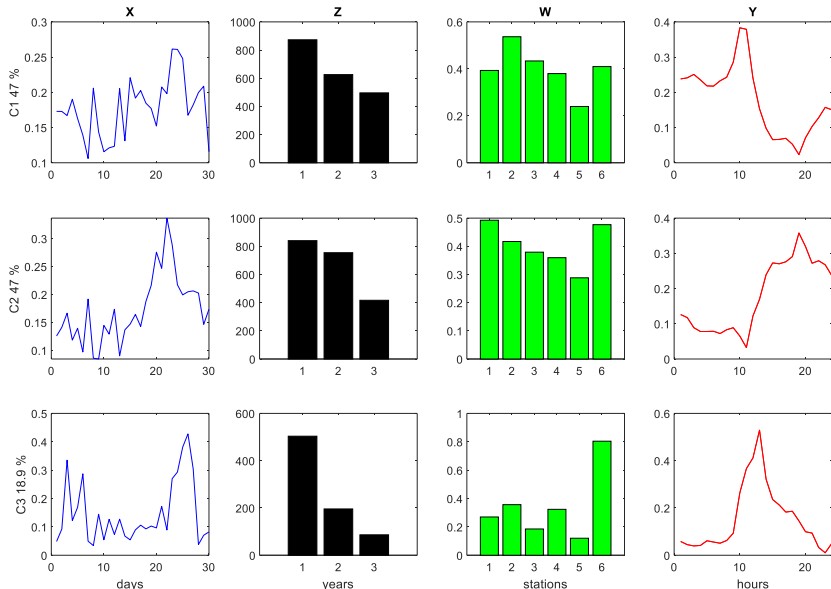

**Figure 6. MCR-ALS analysis of PM$_{10}$ concentrations in the column-wise super-augmented data matrix D$_{caug-}$**
**$_{April-PM10}$ (Eq. (3)) using non-negativity and quadrilinearity constraints. Profiles of the four different data modes**
**are given in different colors: (X) in blue days of April; (Z) in black year 1=2018, 2=2019 and 3= 2020; (W) in**
**green stations 1: Vall d'Hebron, 2: Granollers, 3: Manlleu, 4: Juneda, 5: Bellver, 6: Gràcia; and (Y) in red,**
**hours of the day.**

### 3.1.4. NO$_2$, O$_3$ and PM$_{10}$ simultaneous study (DA12$_{craug}$ data matrix)

MCR-ALS resolved profiles of the **DA12$_{craug}$** data matrix (see Method Sect. 2.7) are given in Fig. 7. Results obtained for the hour profiles (y-mode, in red) for the three pollutants, NO$_2$, O$_3$ and PM$_{10}$, are overlaid in the same plot with the same hour time axis. In this way, possible correlations among the different pollutants can be better explored in these plots. Profiles of components 1 (C1) and 2 (C2) mostly described the O$_3$ pollution: C1 hour profile showed an ozone day-time profile with a wide maximum between 12:00 and 22:00h and C2 described the ozone night-time profile, again with a large maximum between 00:00 and 10:00h. Component 3 described both PM$_{10}$ and NO$_2$ correlated pollution sources, with particulate matter having the highest contribution. The correlation between NO$_2$ and PM$_{10}$ can be due to the common sources of these contaminants (*i.e.*, traffic and industry). Component 4 described the night-time profile of NO$_2$ and last component 5 showed the daily NO$_2$ profile with two maxima, one in the morning (10:00-15:00h) and another at the late evening (20:00- 22:00h), probably associated to the traffic. From the year profiles in z-mode, the evolution of the pollution in the month of April along 2018, 2019 and 2020 could be elucidated. Interestingly, for components 1 and 2 (mostly describing O$_3$ pollution), the variation remained rather constant for the month of April during these three years. Moreover, the variation among stations in the profiles (w-mode) for the two first components was very little. Only in component 2, the stations of Begur and Observatori Fabra showed a higher O$_3$ contribution, probably due to the lower titration effect produced in rural areas and in the Observatori Fabra control site, as previously observed in the individual MCR-ALS



analysis of $O_3$ data. In contrast, the variation among stations and among years was more significant for the
rest of components (C3-C5), mainly describing $NO_2$ and $PM_{10}$ contamination. As observed in z-modes,
contamination by $NO_2$ and $PM_{10}$ was lower in 2019 and even lower in 2020. Considering that the most
important source of $NO_2$ is traffic, the decrease in 2019 can be explained by the implementation of the low
emission zones (LEZs(LEZ - Àrea Metropolitana de Barcelona, 2020)) in Barcelona, as a traffic restriction
policy, first implemented on 2017 and finally put into permanent effect on January 1, 2020. However, the
decrease observed in 2020 might be mostly associated to the COVID-19 lockdown restrictions, being April
2020, the time when the strictest confinement was declared in Catalonia (Real Decreto-ley 10/2020, de 29
de marzo, por el que se regula un permiso retribuido recuperable para las personas trabajadoras por cuenta
ajena que no presten servicios esenciales, con el fin de reducir la movilidad de la población en el contexto
de la l, 2020; Real Decreto 463/2020de 14 de marzo, por el que se declara el estado de alarma para la
gestión de la situación de crisis sanitaria ocasionada por el COVID-19., 2020). Regarding the variation
among stations, components C3 to C5 showed a higher $NO_2$ and $PM_{10}$ contribution in three urban stations
(Vall d'Hebron, Granollers and Gràcia), which is in accordance to the higher traffic density registered on
these sites. The results observed for components C3 to C5 regarding $NO_2$ and $PM_{10}$ pollution were in
concordance to those of their respective individual models evidencing the good performance of the MCR-
ALS simultaneous analysis of the incomplete multiblock data sets and the confirmation of the reliability of
the proposed approach.

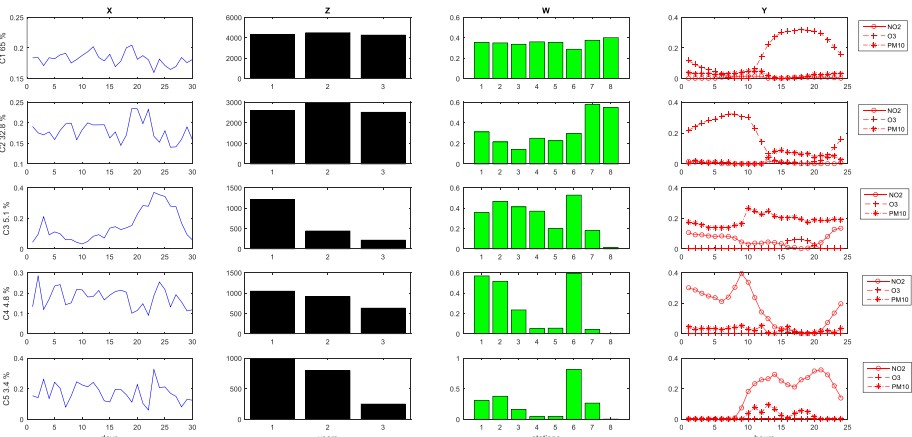


**Figure 7. MCR-ALS analysis of $NO_2$, $O_3$ and $PM_{10}$ concentrations in the column-wise super-augmented**
**incomplete April data matrix $DA12_{craug}$ (see Eq. (9)) using non-negativity and quadrilinearity constraints.**
**Profiles of the four different data modes are given in different colors: (X) in blue days of the year; (Z) in black**
**year 1=2018, 2=2019 and 3= 2020; (W) in green stations 1: Vall d'Hebron, 2: Granollers, 3: Manlleu, 4: Juneda,**
**5: Bellver, 6: Gràcia, 7: Begur, 8: Observatori Fabra; and (Y) in red, hours of the day.**
**3.2. Study of the entire years**
**3.2.1. $NO_2$ study ($D_{caug-allyear-NO2}$ data matrix)**



Profiles obtained by MCR-ALS using non-negativity and quadrilinearity constraints are shown in Fig. S4.
The hour profiles of the 4 resolved components in the analysis of the entire year were similar to those
obtained in the analysis of the month of April: C1 hour profile in April's model was equivalent to C3 hour
profile in all years' model and C2 and C4 hour profiles were equivalent in both models. Also, the diminution
observed in z-mode profile in 2019 and in a bigger extent in 2020 in the month of April was also produced
when analyzing all the year, but in a lesser extent. This might be due to the fact that the traffic restriction
policies were mostly implemented during the strictest confinement (from March 14th to May 4th in
Catalonia) and were gradually removed in the de-escalation phases(Gorrochategui et al., 2021). Also, the
extraordinary rainy conditions registered in April 2020 (Gorrochategui et al., 2021) were not registered for
the rest of the months, making the $NO_2$ depletion less noticeable in the analysis of the whole year. Regarding
stations, the ones showing the higher contribution were the same three urban stations (*i.e.*, Vall d'Hebron,
Granollers and Gràcia) observed in the study of the month of April. Observe also that in the day-of-year x-
profiles some seasonal tendencies can be observed in C1 and C2, with their lower intensities in the middle
of the profile corresponding to the warmer seasons with higher sunlight radiation and higher $NO_2$ depletion
due to the photochemical reaction to form $O_3$. C3 and C4 year profiles did not show major differences over
the year.

**3.2.2. $O_3$ study ($D_{caug-allyear-O3}$ data matrix)**


Only few differences between the analysis of the entire years versus that of the month of April were
observed in the inter-year z-mode (Fig. S5). Component 2 in all years' model, corresponding to a late
evening peak of $O_3$, suffered a slightly significant increase in 2019 and 2020 respect to 2018, which was
not observed in the analysis of the month of April. Such increment can be explained by the reduction of the
titration effect, which was a little higher when considering all the year. The diminution of C3 in 2019 and
in 2020 was also more evident when analyzing the entire years. In this case, this component was associated
to the daily maximum of $O_3$, coincident with the sunlight hours and summer and spring seasons, when the
photochemical reactions with $NO_X$ take place to form ozone. The reason why the changes in $O_3$ were more
evident when considering all year instead of when analyzing just the month of April, opposed to what
happened with $NO_2$, might be due to the fact that despite the traffic restrictions were gradually removed in
the de-escalation phases, the curfew policies remained, causing a potential cumulative suppression of the
titration effect.

**3.2.3. $PM_{10}$ study ($D_{caug-allyear-PM10}$ data matrix)**


As occurred with $NO_2$ and $O_3$, the profiles of the components in $PM_{10}$ MCR-ALS analysis of all year were
similar to those obtained in the analysis of the month of April (Fig. S6). C1 and C3 hours profile in April's
model were equivalent to C3 and C1 in all years' model, respectively, and C2 described the same $PM_{10}$
profile in both models. Also, the diminution observed in 2019 and in a bigger extent in 2020 in the month
of April was also produced when analyzing all the year, but in a lesser extent, as stated for $NO_2$. Moreover,
the meteorological stations with higher contribution in the model were the same than in the model of April,
except for Manlleu, which showed a significant contribution in C2 of this model for the first time when the
entire year $PM_{10}$ data were investigated.




**3.2.4. NO$_2$, O$_3$ and PM$_{10}$ simultaneous study (DY12$_{craug}$ data matrix)**
MCR-ALS resolved profiles of **DY12$_{craug}$** are given in Fig. 8. As observed in the y-mode profiles,
components 1 and 2 mostly described O$_3$ pollution: C1 showed an ozone profile with little daily variation
whereas C2 described a wide O$_3$ maximum between 14:00 and 20:00h. Moreover, the seasonal trend of C2
(x-mode) showed a wide maximum coincident with the solar radiation registered in summer and spring
months. C2 was higher in the urban stations and lower in the rural station of Begur, which could indicate
that such O$_3$ resulted from the photochemical reaction among NO$_x$ in the presence of sunlight in highly
transited areas. Component 3 clearly showed the nighttime profile of O$_3$, with a wide maxim between 17:00
and 00:00h. Interestingly, this component was the only one showing clearly an increase in 2019 and 2020
respect to 2018. As explained in the individual model, such increase is due to the diminution of the titration
effect. Component 4 described the NO$_2$ profile, with a first maximum between 09:00 and 12:00h and a
second but lower maximum at late evening (20:00-00:00h). Component 5 described the simultaneous
contribution of NO$_2$ and PM$_{10}$, with higher contribution of PM$_{10}$, having again the same two-maxima profile
observed in component 5 for NO$_2$. Interestingly, both components 5 and 6 presented maximums in the
urban stations (Vall d'Hebron, Granollers and Gràcia) and a decrease in 2020, due to the traffic diminution
registered during the COVID-19 lockdown.

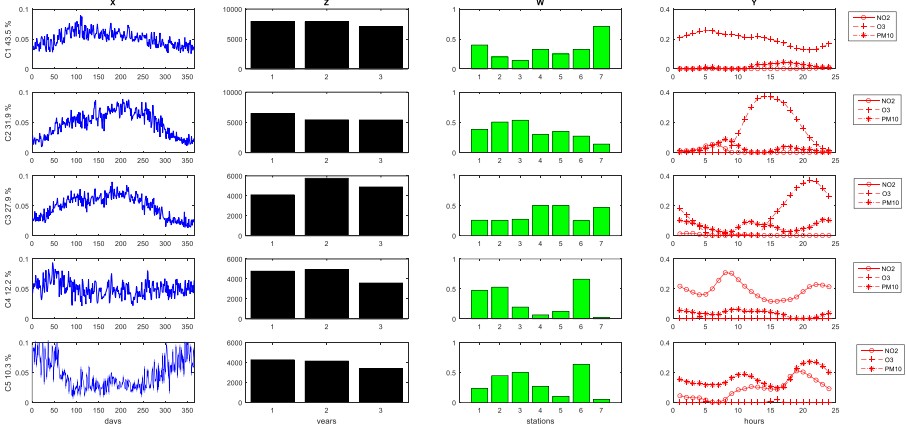


**Figure 8. MCR-ALS results of the simultaneous analysis of NO$_2$, O$_3$ and PM$_{10}$ for the entire years (incomplete**
**super-augmented data matrix D$_{Y12craug}$) using non-negativity and quadrilinearity constraints. Profiles of the**
**four different data modes are given in different colors: days of the month April (X) in blue, year (Z) in black,**
**stations (W) in green and hours (Y) in red. In the y-mode, the hourly profiles of the three contaminants are**
**overlapped. Capital letters in the figure indicate the different stations: 1: Vall d'Hebron, 2: Granollers, 3:**
**Manlleu, 4: Juneda, 5: Bellver, 6: Gràcia, 7: Begur.**
**4. Conclusions**
MCR-ALS with quadrilinearity constraints has demonstrated to be a powerful tool to resolve the principal
contamination profiles of four-way environmental datasets, even when containing missing data blocks. The
main advantage provided by the use of quadrilinearity constraints is the better and easier interpretability of
the profiles, which appear more condensed and concise.



In this study, resolved MCR profiles using quadrilinearity constraints have been shown to describe
adequately the different patterns and evolution of $NO_2$, $O_3$ and $PM_{10}$ contamination during the different
hours of the day, during the different days (hourly and daily variations) for the two periods of time
evaluated: month of April versus the entire year for 2018, 2019 and 2020. For each period of time studied,
the individual models of the contaminants together with their simultaneous analysis have been performed.
The simultaneous analysis of the incomplete multiblock data sets allowed the exploration of the potential
correlations among the three contaminants, which was easily interpretable with the representation of
overlapped $NO_2$, $O_3$ and $PM_{10}$ hourly profiles. Interestingly, both in the study of the month of April and the
study of the entire years, the simultaneous analysis of the three contaminants evidenced a correlation
between $NO_2$ and $PM_{10}$, due to their common pollution sources (*i.e.*, traffic and industry). Moreover, the
profiles of these two contaminants showed an inter-year decrease, due to the introduction of LEZs (LEZ -
Àrea Metropolitana de Barcelona, 2020) in 2019 and due to the COVID-19 lockdown restrictions and to
the high amount of rainfall registered in April 2020(Real Decreto 463/2020de 14 de marzo, por el que se
declara el estado de alarma para la gestión de la situación de crisis sanitaria ocasionada por el COVID-19.,
2020). Such decrease was consistently higher in the three most transited urban stations studied: Vall
d'Hebron, Granollers and Gràcia.
On the other hand, MCR-ALS ozone profiles both in individual and simultaneous models presented an
opposite inter-year trend, especially when analyzing the entire years. Globally, $O_3$ profiles showed an
increase in 2019 and in 2020 respect to 2018, which can be attributed to the diminution of the titration effect
linked to the lockdown and curfew restrictions. Such effect was more evident in inner rural areas and in the
control site (*i.e.*, Begur and Observatori Fabra), where the amount of $NO_x$ necessary to react with ozone
and to produce its suppression is lower compared to urban areas due to the smaller traffic density and
industrial activity.
Overall, this work contributes to the better knowledge of the evolution of $NO_2$, $O_3$ and $PM_{10}$ contamination
in eight rural and urban areas of Catalonia during the two years before the COVID-19 (*i.e.*, 2018 and 2019)
and the year itself of the pandemic (*i.e.*, 2020). The work also highlights: (a) the capacity of MCR-ALS
with quadrilinearity constraints to perform simultaneous analysis of different contamination sources to
study potential correlations among them and (b) the good performance of this approach in the analysis of
complex four-way environmental data sets containing missing data blocks, providing concise and easy
interpretable results.

**Author contributions**
EGM performed data curation, formal analysis and writing. IH provided air quality data. RT contributed to
data curation and global supervision.

**Competing interests**
The contact author has declared that neither they nor their co-authors have any competing interests.
**Financial support**





This study was supported by the Ministry of Science and Innovation of Spain under the project PID2019-
105732GB-C21.

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
