# Peer review of "A model for simultaneous evaluation of NO2, O3 and PM10"

_EGUsphere, 2022_

## Author Response (AR1)

R1: This study attempts to apply multivariate curve resolution alternating least squares with quadrilinearity constraints to handle complex and incomplete four-way atmospheric data sets. Three air pollutants including nitrogen dioxide, ozone and particular matter in eight sampling stations located in Barcelona metropolitan area other parts of Catalonia during the COVID-19 lockdown (2020) with respect to previous years (2018 and 2019) are used for analysis. The new method indeed generates an interesting result, which has been well interpreted. Overall, the manuscript is well written and worthy of publication. However, this reviewer disagrees on one issue, i.e., estimation of missing data always suffered from uncertainties, whatever any approach to be used. The uncertainty should be included in data interpretation.

We thank the reviewer for the comment. And yes, we agree that measurement uncertainties were not included in the bilinear model factor decomposition estimations. The environmental agency source data did not provide them. Otherwise, we could have applied the weighted version of ALS where data uncertainties are included as weights in the least squares estimations. On the other hand, missing data blocks were not included in the least squares estimations, this is the advantage of the proposed method, linear equations were only solved for the known data blocks. Therefore, this should not be a limitation. What is true is that some parts of the factor solutions (those corresponding to the missing blocks) are not so overdetermined from a least squares point of view as the other data blocks without missing values, and this can be reflected in the reliability of the estimations of the later. This is an aspect that deserves a deeper study and needs further investigation.

R2: The present study proves convincingly the advatage of the chemometric technique multivariate curve resolution - alternating lesdt squares. The reasons for such an estimations could be summarized as follows:

1. Reliable options for apportionment of pollution sources

2. Realing with four-way constructed data sets

3. Elimination of missing data

The study is performed on a very high theoretical level and, additionally, delivers very useful practical information. The style is very sound, the conclusions - important.

I recommend acceptance in its present form.

We kindly appreciate this reviewer's comment. We also agree that this study highlights the advantages of chemometrics in the analysis of complex four-way environmental data sets containing missing data blocks.